# Inadequacy of fluvial energetics for describing gravity current autosuspension

**Sojiro Fukuda** [ORCID][1] [✉], **Marijke G. W. de Vet** [ORCID][1], **Edward W. G. Skevington** [ORCID][1], **Elena Bastianon**[1], **Roberto Fernández** [ORCID][1], **Xuxu Wu**[1], **William D. McCaffrey**[2], **Hajime Naruse** [ORCID][3], **Daniel R. Parsons** [ORCID][1] & **Robert M. Dorrell** [ORCID][1]

Gravity currents, such as sediment-laden turbidity currents, are ubiquitous natural flows that are driven by a density difference. Turbidity currents have provided vital motivation to advance understanding of this class of flows because their enigmatic long run-out and driving mechanisms are not properly understood. Extant models assume that material transport by gravity currents is dynamically similar to fluvial flows. Here, empirical research from different types of particle-driven gravity currents is integrated with our experimental data, to show that material transport is fundamentally different from fluvial systems. Contrary to current theory, buoyancy production is shown to have a non-linear dependence on available flow power, indicating an underestimation of the total kinetic energy lost from the mean flow. A revised energy budget directly implies that the mixing efficiency of gravity currents is enhanced.

"Consider the [turbidity] current as … a river" R. A. Bagnold (1962); the foundation of contemporary deep marine sedimentology.

Gravity currents are a broad class of flows with a wide range of environmental applications, including terrestrial cold fronts and submarine thermohaline currents[1]. Of particular interest are particle-driven gravity currents, such as powder snow avalanches, pyroclastic density, and turbidity currents. Turbidity currents have received significant attention, due to: their capacity to travel long distances, 100s-1000s of kilometres, along sinuous submarine canyon-channel systems[2,3]; their importance to the deep marine environment[4,5]; the depositional record of paleoenvironments[6]; and for geohazard risk management[7–9].

Turbidity currents are generated by the presence of suspended sediment, meaning they have a higher density than ambient water. This density difference generates a downslope gravitational force. The resulting flow produces turbulent mixing, keeping sediment in suspension[10]. This suspension-flow feedback loop is referred to as autosuspension, the minimal requirement for long runout[11,12] in all particle-driven gravity currents. Accurate prediction of autosuspension is essential to quantify gravity current propagation, natural hazard risk, and, for turbidity currents, deep marine biogeochemical cycling and anthropogenic environmental impact. However, despite its importance, the mechanisms that enable autosuspension are poorly understood because the kinetic energy of the flow is consumed to maintain the particles in suspension, and uplift them during turbulent mixing with the environment[3], which ultimately stalls the flow.

Historically, autosuspension has been explained by the positive feedback whereby sediment entrainment increases the turbulence, referred to as self-acceleration[10]. Where the slope is steep, such as in the proximal regions of submarine canyons or on volcanic slopes, gravitational forcing (proportional to the slope) is relatively large and thus gravity currents are predominantly net-erosional. Entrainment of sediment provides the flow with additional mass and driving force, increasing the momentum and the basal drag, which in turn increases the turbulent energy, further enhancing sediment entrainment and accelerating the flow[10]. However, turbidity currents can propagate for extensive distances and, in distal reaches, they can traverse near-zero slopes ($< 10^{-4}$ m/m)[13]. In these regions, the gravitational forcing, proportional to slope, is small. Consequently, the velocity is reduced and the flow is, at best, only weakly erosional, if not net-depositional. Without net-sediment erosion, the work done due to the entrainment of the ambient fluid decelerates the flow[10], and

[1]Energy and Environment Institute, University of Hull, Hull, UK. [2]School of Earth and Environment, University of Leeds, Leeds, UK. [3]Department of Geology and Mineralogy, Division of Earth and Planetary Sciences, Graduate School of Science, Kyoto University, Kyoto, Japan. ✉e-mail: S.Fukuda-2018@hull.ac.uk

ultimately causes sediment deposition. The loss of sediment reduces the driving force and decelerates the flow. This negative feedback loop stalls the flow, precluding self-acceleration as an explanation of auto-suspension. For partially confined flows, including channel-levee systems, this is exacerbated by the loss of mass and momentum due to overspill[14].

The turbulent kinetic energy (TKE) of a flow is often taken as a measure of its capability to suspend sediment[11,12,15]. While there may be local regions where the TKE is dissipated by the flow dynamics, over the total length of the flow the TKE generated and dissipated is vastly greater than the amount stored at any one location. Consequently, the net advection of TKE is negligible[16], and a necessary condition for autosuspension is that the energy loss of the mean flow integrated over the bed-normal direction, $P_{loss}$, must exceed the integrated buoyancy production required to maintain the sediment in suspension, $B_{gain}$, and the integrated viscous dissipation, $\varepsilon$. This yields the famed Knapp–Bagnold (K-B) autosuspension criterion[11,12]

$$P_{loss} > B_{gain} + \varepsilon. \tag{1}$$

Bagnold[12] proposed that energy is lost by the mean flow through the shear production of turbulence, and gained by the sediment through turbulent uplift, and this assumption has been widely adopted by subsequent authors[10,17–23]. Explicitly, the assumption is that, to leading order,

$$P_{loss} \simeq P_{shear} \equiv \int_0^h -\langle u'w'\rangle \frac{\partial \langle u\rangle}{\partial z} \, dz, \quad \text{and} \quad B_{gain} \simeq B_{turb} \equiv \int_0^h gR\langle w'\phi'\rangle \, dz. \tag{2}$$

Here and throughout, $\phi$ and $\Phi$ denote local and depth-averaged volumetric sediment concentration respectively; $u$ and $U$ denote the local and depth-averaged fluid velocity; $g$ denotes gravitational acceleration; $R = \rho_s/\rho - 1$ denotes reduced density; $w_s$ denotes particle settling velocity; $h$ is the extent of the current in the bed normal direction $z$, i.e. the flow depth. Primes and angled brackets denote the Reynolds fluctuations and time-averaged values respectively.

For equilibrium fluvial flows, $\langle w'\phi'\rangle = \phi w_s$[24], such that $B_{gain} \simeq B_f \equiv gR\Phi h w_s$. Moreover, assuming a logarithmic velocity profile, the energy loss in fluvial flow is estimated as $P_{loss} \simeq P_f = u_*^2 U$, where $u_*$ is the shear velocity[10,12,23,25,26]. Indeed, equilibrium fluvial flow models for suspended sediment transport, based on the same physical arguments as the K-B criterion[15,27], provide a extensively validated linear proportionality between the concentration of suspended sediment and the dimensionless flow power[20,22,28,29],

$$\Phi \propto \frac{P_f}{N_f} = \frac{u_*^2 U}{gRh w_s}, \tag{3}$$

where $N_f = B_f/\Phi$ is the normalised buoyancy production term (the energy required to suspend a unit volume of sediment).

A common class of closures for flows are 'top-hat' models, where there is no vertical variation in flow structure. However, via basal shear, these models do capture log-law production energetics to leading order, and the two approaches are equivalent for fluvial systems[30]. Such models have been extended to gravity currents[31], including turbidity currents[10], and form the basis of contemporary system scale models[10,14,19,21]. The dimensionless flow power inherent to top-hat models of gravity currents can be derived from the kinetic energy conservation equation of the mean flow, which is only modified by the presence of entrainment

$$\frac{P_{th}}{N_{th}} = \frac{u_*^2 U + \frac{1}{2}e_w U^3}{gRh(w_s + \frac{1}{2}e_w U)}. \tag{4}$$

Here $P_{th}$ is derived from the 'top-hat' gravity current model as the energy loss from the mean flow, $P_{loss}$, and it is assumed that all energy lost is attributed to the shear production of TKE. Moreover, $B_{th} = \Phi N_{th} = \Phi gRh(w_s + \frac{1}{2}e_w U)$ is the buoyancy production in the model, where $e_w$ is the water entrainment rate, calculated as the energy required for the sediment to remain uplifted, $B_{gain}$, and assumed equal to the turbulent uplift. If there is no entrainment, $P_{th} = P_f$ and $B_{th} = B_f$, thus these are the minimal adjustments to fluvial theory to include entrainment. Consequently, for top-hat models to be valid, it is required that the turbulence in gravity currents is essentially the same as in fluvial systems, despite the substantial differences in the flow structure. This implicit assumption is the target of the present analysis.

In this work, the correlation between the total energy loss of a mean flow, $P_{loss}$, and the energy required to keep sediment in suspension, $B_{gain}$, is reviewed for near-equilibrium flows, to investigate whether idealised flow power theory is an appropriate predictor for autosuspension. While the total energy loss of the mean flow in gravity currents is unknown, it has previously been assumed[10,12,23,25,26] proportional to the log-law total-shear TKE production, $P_f$ or the mean-flow energy loss predicted in the top-hat model, $P_{th}$. This study reviews experimental and direct observation of gravity currents available in the literature, adding new experiments to directly address data gaps. Crucially, data shows that the total energy loss of the mean flow, $P_{th}$, has a non-linear dependence on the work required to keep sediment in suspension, $B_{th}$. A review of the energy deficit implies that particulate transport in gravity currents is driven by mixing at scales larger than that of TKE.

## Results

To parameterise energy balance for equilibrium flows in autosuspension, new experiments have been conducted and integrated with over 70 years of empirical and observational data of turbidity currents and dynamically similar particle-driven pyroclastic density currents. Figure 1 highlights that this dataset uniformly spans a wide range of flow states, from subcritical to supercritical with small to large drag coefficients. Also included are all laboratory-scale studies of constant-discharge sediment-laden turbidity currents in straight channels (see Supplementary Table 1 and Supplementary Note 2 for more details). The experiments of this study are designed to address extant data gaps, which have limited the full understanding of autosuspension and gravity currents dynamics (see 'Methods' and Supplementary Note 3). Data are separated into types that have both velocity and concentration profiles (TYPE I) and types where concentration is estimated from inlet conditions (TYPE II).

The strict requirement for no net sediment deposition is used to limit data parameterising the energy balance of autosuspension. When the bed shear stress, described by the dimensionless Shields number, $\tau_* = u_*^2/gRd_{50}$ where $d_{50}$ is the median particle size, is less than the threshold needed for incipient sediment motion no sediment can be maintained in suspension: the flow is depositional. In Fig. 2 the criterion of Guo[32] (solid curve) is taken as the minimal $\tau_*$ for incipient motion. Thus none of the TYPE II data and 19% of TYPE I data (38 out of 203 points) are excluded as belonging to strictly dispositional flow. The remaining turbidity current data lies within the suspended load regime for dilute flows[27].

### Sediment transport capacity

In equilibrium flows, the total kinetic energy loss of the mean flow, $P_{loss}$, is assumed to be well approximated by simplified 'top-hat' models. The energy loss balances both the work done to keep sediment in suspension and viscous dissipation, Eq. (1). When TKE production is dominated by the effects of basal drag then the energy available to uplift sediment is given by the log-law of the wall, $P_{loss} \simeq P_f$. Thereby, a linear correlation is implied between the volumetric

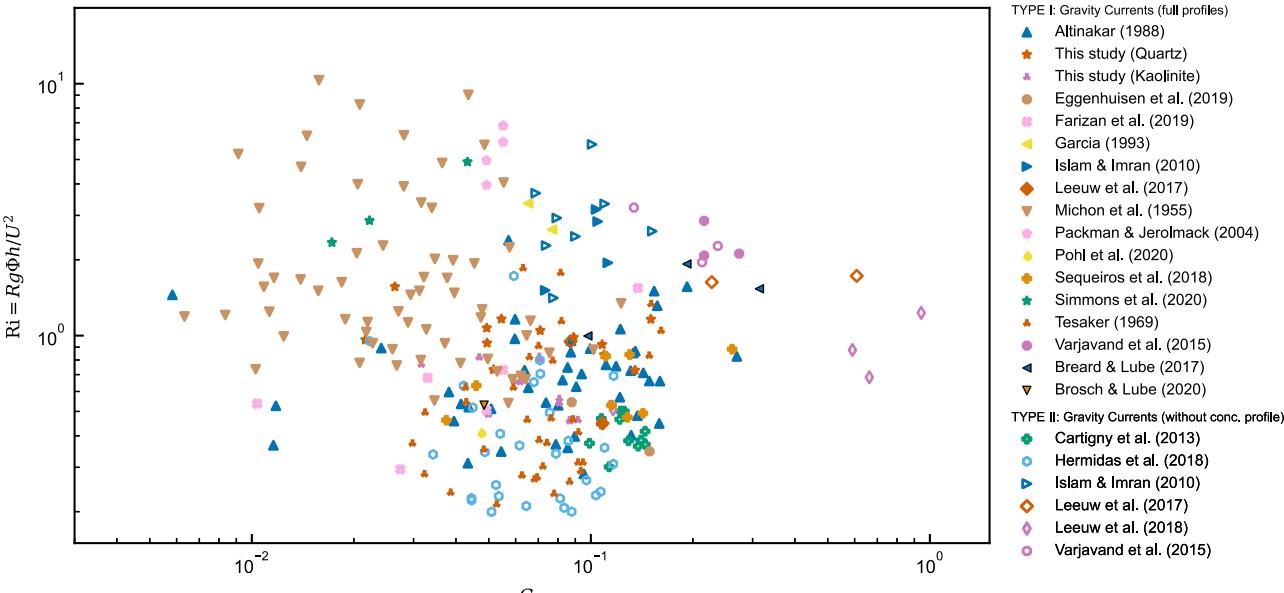

**Fig. 1 | Data distribution in parameter space.** Drag coefficient, $C_D$, and bulk Richardson number, $\text{Ri} = gR\Phi h/U^2$, of the compiled gravity current data. Symbols depict field observations and experimental studies of particle-driven gravity currents (See Supplementary Table 1 for the detailed reference and setting of experiments in each source). Black borders denoted pyroclastic density current experimental data.

concentration and dimensionless flow power, $P_f/N_f$, where $N_f = gRhw_s$. This flow-power balance is tested against compiled near-equilibrium laboratory- and natural-scale gravity currents (Fig. 3 and Supplementary Note 4 and 5).

From the fluvial flow data (Fig. 3a), three regimes of concentration are identified: dilute, transitional, and hyperconcentrated flow. Dilute flows, $\Phi \lesssim 10^{-2}$, are characterised by a linear increase in concentration with dimensionless flow power, $P_f/N_f$ (Fig. 3a and Table 1(i)). With increasing concentration, $10^{-2} \lesssim \Phi \lesssim 10^{-1}$, transitional flows exhibit a change in correlation from increasing to decreasing dimensionless

flow power as concentration increases. This transition may be explained by the onset of turbulence dampening and particle-particle interactions dominating the sediment transport mechanics[29]. Hyperconcentrated flows, $\Phi \gtrsim 10^{-1}$, are characterised by a non-linear decrease in dimensionless flow power with increasing concentration. As concentration continues to rise, dimensionless flow power decreases by at least two orders of magnitude.

The gravity current data (Fig. 3b–e) also exhibit three similar regimes of concentration; the threshold concentrations between regimes are approximately equal to those of fluvial systems. However, the correlation of concentration with dimensionless flow power is remarkably different. The dimensionless flow power, based on both the log-law energy production model $P_f$ (Fig. 3b), and the top-hat model, $P_{th}$ (Fig. 3c), have a strongly non-linear correlation with sediment concentration, $\Phi$. The fit of both models is good, but shows an improvement when using the top-hat-based correlation, see Table 1 ((ii) and (iv)). However, a linear model provides a poor best fit in comparison, contrast Table 1((iii) and (iv)). Moreover, the turbidity current data (Fig. 3d) show almost identical non-linear dependency to the pyroclastic density current data (Fig. 3e), suggesting that the non-linear dependency is universal to all types of gravity currents.

Critically, the non-linear relationship results in dilute gravity currents being able to maintain a higher suspended sediment concentration versus fluvial flows of an equivalent dimensionless flow power (Fig. 3c). Previously unrecognised, this has the potential to explain autosuspension in long-runout systems. The correlation suggests that when a dilute gravity current accelerates, it is not as erosive as a fluvial system, and similarly, when a gravity current decelerates, deposition is more limited. This implies that the suspended-load of gravity currents is significantly underestimated, i.e. providing more motive force on shallower slopes, and is less sensitive to changes in flow power than has previously been assumed based on the use of fluvial analogues[10,17–23]. Since the limited super-dilute pyroclastic density current dataset also exhibits a similar non-linear trend to turbidity currents (Fig. 3e), it is likely that the fluvial-based or top-hat gravity current models are a poor approximation not only for turbidity currents but also for particle-driven gravity currents in general.

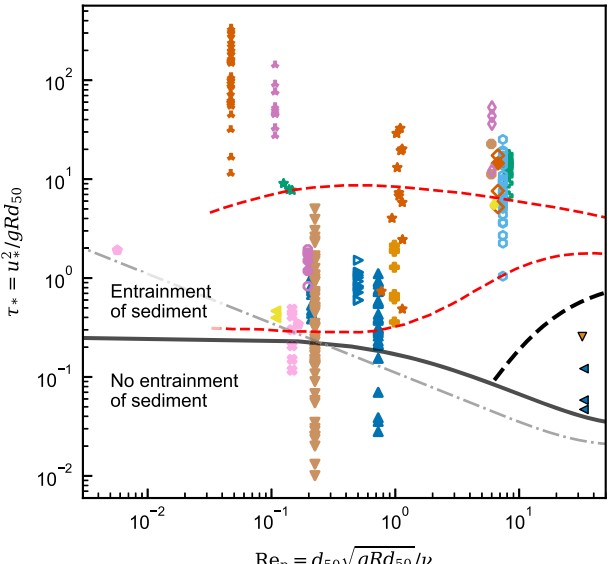

**Fig. 2 | Shields diagram.** Shields number, $\tau_*$, versus the particle Reynolds number, $\text{Re}_p$, of experimental and observational gravity current data (Fig. 1). Here, $\nu$ denotes the kinematic fluid viscosity. Solid[32] and dash-dot[60] curves depict the criteria of incipient motion. The black dashed line depicts the $w_s = u_*$ criterion for suspended load[27]. Equilibrium sediment suspension criteria for monodisperse and poorly-sorted suspensions are depicted by the lower and upper red dashed curves respectively[22]. The symbols and colours are as per Fig. 1.

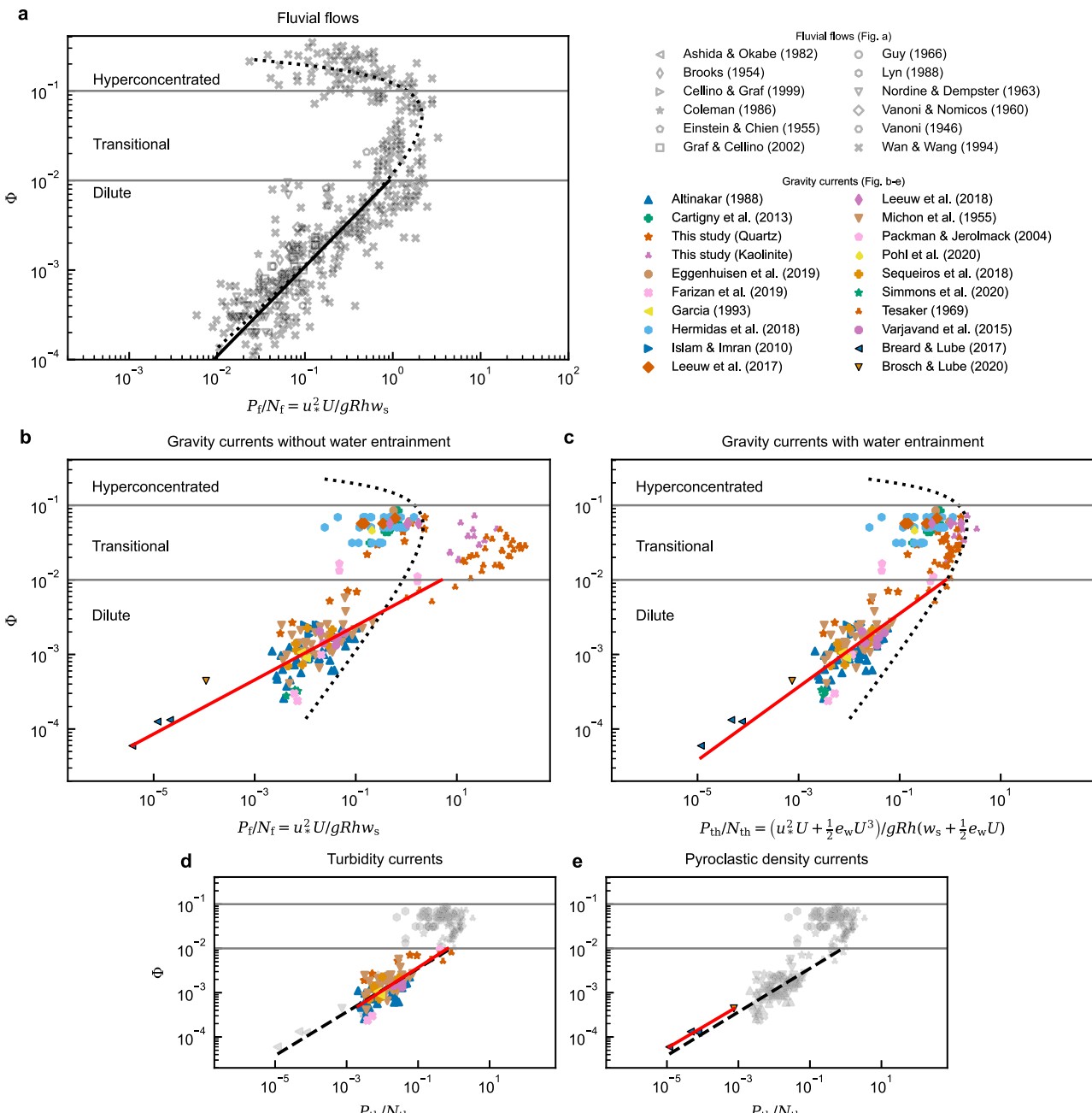

**Fig. 3 | Sediment transport capacity for fluvial flows and gravity currents.** Sediment concentration, $\Phi$, versus dimensionless flow power, $P_-/N_-$ for: **a** fluvial data (see supplementary material for the detailed reference of each source) with the log-law model, $P_f/N_f$; **b** all gravity current data with the log-law model; **c** all gravity current data with the top-hat, entrainment based, model, $P_{th}/N_{th}$; **d** turbidity current concentration with the top-hat model; and **e** pyroclastic density current concentration with the top-hat model. Dilute, transitional and hyperconcentrated regimes are separated by grey solid lines. The parametric correlation of concentration and dimensionless flow power in fluvial systems (fitted using an iterative least squares method, see supplementary material) is depicted by a black dotted curve. Power-law correlations (Table 1) in dilute regimes of each subset are depicted by black (**a**) and red (**b**–**e**) solid lines. Black dashed lines (**d**–**e**) represent the power-law correlation of dilute gravity currents (red solid line in **c**).

It is noted that a limited proportion of the data (see Supplementary Table 1) use cohesive material (kaolinite). It is plausible that flocculation of cohesive particles may occur, increasing particle settling velocity, and decreasing dimensionless flow power. However, the growth of flocs is limited by shear rates and their size decreases as the flow increases[33]. Therefore, in strongly sheared gravity current experiments, the development of flocs and resulting underestimation of settling velocity is expected to be limited. Further, it would not change the observation that the relationship between concentration and dimensionless flow power is non-linear, see Fig. 3b–e, Table 1((ii) and (iv)). Moreover, while the increase in $P_{th}/N_{th}$ of gravity currents (Fig. 3c) follows the trend of fluvial data in the transitional regime it is based on an empirical water entrainment function, $e_w$. The empirical water entrainment function has been developed for dilute currents, thus it is expected the values of $P_{th}/N_{th}$ in the transitional regime have some inherent error. However, this does not impact the primary findings of the non-linear correlation in the dilute regime.

## Discussion

The fundamental differences between fluvial flows and dilute gravity currents (turbidity currents and pyroclastic density currents) documented above raise the following questions: Do the top-hat mean-flow energy loss, buoyancy production and dissipation balance? What are the implications for autosuspension models?

In top-hat gravity current models[10], $P_{th}$ is the energy released by the mean-flow and available for the suspension of particles, which requires energy $B_{th}$. Additionally, the flow experiences viscous dissipation of energy $\varepsilon$. However, as will be shown, there is insufficient energy in this model to suspend the sediment. For the present analysis, the missing energy will be denoted by $S$, so that the energy balance is

$$P = P_{th} + S \simeq B_{th} + \varepsilon_{th} = \left(1 + \frac{1}{\Gamma}\right)B_{th}, \quad (5)$$

where the total turbulent flux coefficient, $\Gamma = B_{th}/\varepsilon_{th}$, denotes the ratio of top-hat buoyancy production, $B_{th}$, to dissipation, $\varepsilon_{th}$. Using Eq. (5) and the curve fitting results, see Table 1((i) and (iv)), $\Gamma$ can

**Table 1 | Fitted power-law correlation results and the coefficient of determination, $R^2$, of the subsets of data points (Fig. 3), using orthogonal distance regression (see 'Methods')**

|  | Flow type | Subsets | Figure | Curve fit | $R^2$ |
|---|---|---|---|---|---|
| (i) | Fluvial flows | Dilute | 3a | $1.1 \times 10^{-2} \left(\frac{P_f}{N_f}\right)^{1.0^a}$ | 0.76 |
|  |  | ($P_f$, linear fit) |  |  |  |
| (ii) | Gravity currents | Dilute | 3b | $5.6 \times 10^{-3} \left(\frac{P_f}{N_f}\right)^{0.36}$ | 0.65 |
|  |  | ($P_f$) |  |  |  |
| (iii) |  | Dilute | – | $1.0 \times 10^{-1} \left(\frac{P_{th}}{N_{th}}\right)^{1.0^a}$ | 0.34 |
|  |  | ($P_{th}$, linear fit) |  |  |  |
| (iv) |  | Dilute | 3c | $1.1 \times 10^{-2} \left(\frac{P_{th}}{N_{th}}\right)^{0.49}$ | 0.72 |
|  |  | ($P_{th}$) |  |  |  |

$^a$The power of correlation is fixed as unity for the linear fits.

be expressed as:

$$\Gamma = \left(\frac{P_\sim}{B_\sim} + \frac{S}{B_\sim} - 1\right)^{-1}$$

$$\simeq \begin{cases} 1.1 \times 10^{-2} & : \text{Fluvial flows}\,(S = 0,\ B_\sim = B_f,\ P_\sim = P_f) \\ \left(9 \times 10^3 \Phi^{1.0} + k - 1\right)^{-1} & : \text{Gravity currents}\,(B_\sim = B_{th},\ P_\sim = P_{th}) \end{cases} \quad (6)$$

where $k = S/B_{th}$ denotes the ratio of missing energy to the top-hat buoyancy production, $B_{th}$. The fluvial data (Fig. 3a) and Eq. (6) implies that, for fluvial flows, $\Gamma$ is constant. Only ~1.1% of the energy production is consumed by buoyancy production, while the rest is consumed by dissipation. For gravity currents in contrast $\Gamma$ depends on $\Phi$ (Fig. 4a). Assuming that the extra energy source, $S = 0$ and thus $k = 0$, $\Gamma$ grows with decreasing flow concentration, diverging at $\Phi = 1.4 \times 10^{-4}$, before becoming negative. However, $\varepsilon_\sim$ and $B_\sim$ are strictly positive, thus $\Gamma$ must also always be positive.

Clearly, $S = 0$ is a poor approximation. Eq. (6) implies that the energy balance of near-equilibrium gravity currents can only be satisfied with a non-zero energy source/sink, $S$. To satisfy the minimum requirement, $\Gamma > 0$ for all $\Phi$, the additional energy source term is constrained by $k > 1$. A hypothesised upper limit for the turbulent flux coefficient[16,34] is $\Gamma \leq 0.2$, this is broken for $\Phi < 6 \times 10^{-4}$. To satisfy this limit a value of $k \sim 10$ would be required. However, it is unlikely that turbidity currents reach this maximum mixing efficiency, and a larger value of $k$ is likely required. Previously, it has been assumed that the amount of TKE consumed by buoyancy production in gravity currents is similar to that in fluvial flows[35]. To satisfy $\Gamma \sim 10^{-2}$, Eq. (6), then $k \sim 100$ (Fig. 4b). Thus, K-B type criteria and top-hat gravity current models[10,12,17–23] fail to explain the energy balance of gravity currents.

The energy balance of gravity currents required for autosuspension cannot be explained without an additional energy source, i.e. $S > 0$ in Eq. (5) and Fig. 4. Crucially, if the energetic mechanisms were the same for gravity currents and fluvial flows, then gravity currents would be substantially more dilute, cf. Fig. 3b–e. Therefore, to explain autosuspension, mechanisms for particle uplift must be present that are absent, or of negligible importance, in fluvial flows. It is plausible that the shear production, Eq. (2), predicted by top-hat models is less than the actual production in real flows, and this possibility is addressed first. The shear production is calculated from empirical data[36], and plotted in Fig. 5. However, the limited data available suggest that the

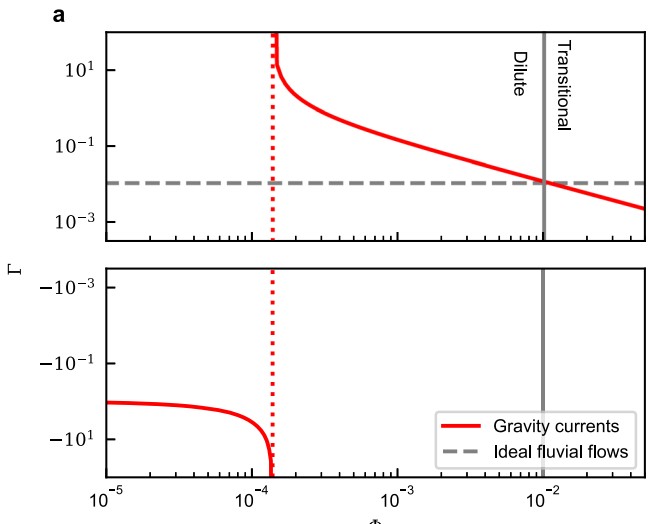

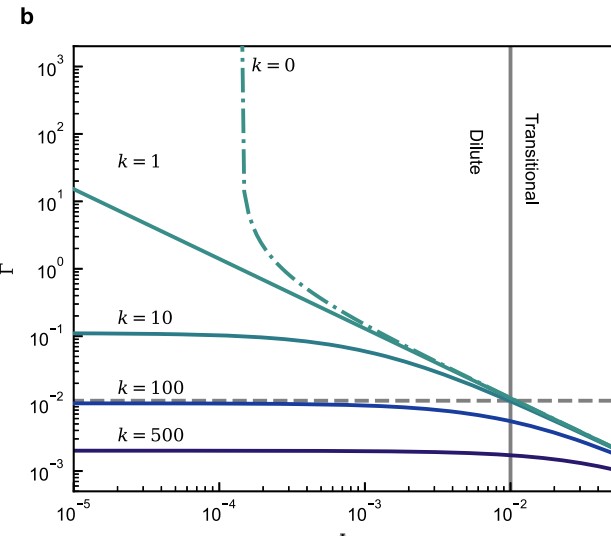

**Fig. 4 | Predicted turbulent flux coefficient.** The turbulent flux coefficient, $\Gamma$, as a function of concentration, $\Phi$. **a** Computed $\Gamma$, assuming $P = P_{th}$ and $S = 0$. The red dotted line indicates the vertical asymptote for the gravity current curve (red solid). **b** Computed $\Gamma$, assuming $P = P_{th} + S$, where the additional energy is parameterised in terms of the buoyancy production $S = kB_{th}$. The dotted-dashed curve has $k < 1$, for which $\Gamma$ diverges to infinity for some values of $\Phi$. Throughout, the Dilute/Transitional regime threshold and the ideal fluvial flow curve is depicted by grey solid and grey dashed lines respectively.

directly computed shear production term, $P_{shear}$, is substantially lower than the 'top-hat' energy loss term, $P_{th}$,

$$\frac{P_{shear}}{N_{th}} = 0.74 \left( \frac{P_{th}}{N_{th}} \right)^{1.27} \quad (R^2 = 0.92). \quad (7)$$

Consequently, if shear production represents all available energy, the size of the missing energy source is substantially larger. To understand the origin of these shortcomings, the energetic dynamics within a gravity current will be broken down and explored conceptually; a schematic of these dynamics is provided in Fig. 6. The dynamics of the flow occur on three distinct length-scales: the macro-scale flow on the scale of the length of the current, which is the scale of the top-hat model; the meso-scale flow on the scale of the depth of the current,

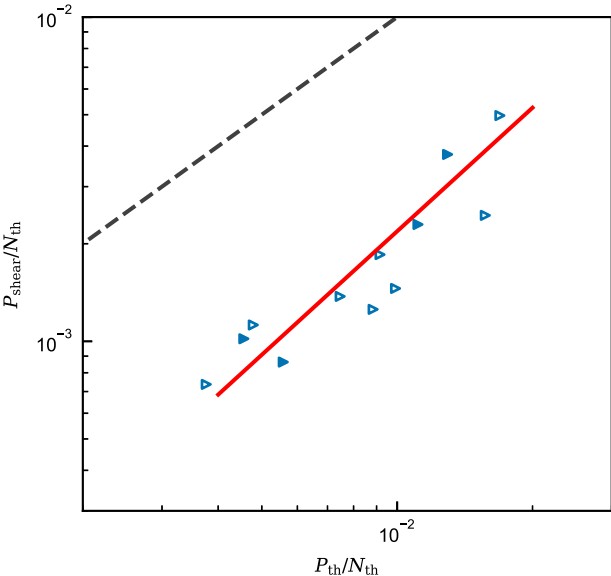

**Fig. 5 | Comparison of different measures of TKE production.** Log-log plot of depth-integrated shear production against top-hat total energy loss of a mean flow $P_{th} = u_*^2 U + \frac{1}{2} e_w U^3$. The power-law correlation (Eq. (7)) is depicted by a red solid line. Grey dashed line indicates the line of equality ($P_{shear} = P_{th}$). Data are calculated from the empirical measurements reported by Islam and Imran[36]. Symbols as per Fig. 1.

which is able to support internal waves[37–41] and the largest vortices[42,43]; and the micro-scale flow, which supports the turbulent vortices (TKE). In Fig. 6 the gravitational potential of the sediment is split between the macro-scale contribution, the vertical distribution of the sediment is (on average) slowly varying, and the meso-scale contribution due to the rise and fall of internal waves, for example.

In simplified models of gravity currents, such as top-hat models, it is the macro-scale kinetic energy that is captured, and on shallow slopes it is this macro-scale flow which is energised by the down-slope component of gravity. As the longitudinal flow accelerates/decelerates, the flow thins/thickens, exchanging the macro-scale energy between the kinetic and gravitational potential (not included in Fig. 6). This kinetic energy is lost at a rate $P_{loss}$. Large scale internal shear can generate flow instabilities, such as the Kelvin–Helmholtz or Holmboe instabilities, resulting in internal waves[37–41], which can be seeded at flow initiation[38,44]. Alternatively, the mean flow energy may be used to stimulate large vortices, for example, secondary flow circulation[42,43,45]. The internal shear generated by the macro-scale flow also directly energises turbulent vortices through shear production. The meso-scale flow structures are able to generate gravitational potential directly, by stirring the flow[45], and through wave breaking which also generates TKE. The turbulence is, in turn, able to 'ring' density interfaces generating internal waves[39,40,46], or uplift particles through the diffusive effect of vortices. The TKE and gravitational potential are slowly lost to heat though viscous effects.

In addition to these internal processes, in many currents external forcing directly drives the flow, also depicted in Fig. 6. Examples include the Coriolis force[47,48], bottom currents[49], tidal forcing, return flows[50–52], and thermal effects in pyroclastic density currents[53,54]. However, these effects are not equally featured in the flows in our dataset, and are expected to result in quite different internal dynamics. Thus, the trend in Fig. 3c and the derived missing energy cannot be explained by these external forces, and instead they generate the scatter about the trend.

It is worth pausing to reconsider what the top-hat model represents. The production, $P_{th}$, is derived as approximating the energy loss from the mean flow $P_{loss}$, and $B_{th}$ as approximating the energy gain by the sediment $B_{gain}$. However, it is clear from Fig. 3 and the implied missing energy that at least one of these approximations is inaccurate, most likely both. Through Fig. 5, we see that $P_{th}$ does not approximate the shear production. Consequently, the top-hat production does not represent *any* of the indicated production terms on Fig. 6.

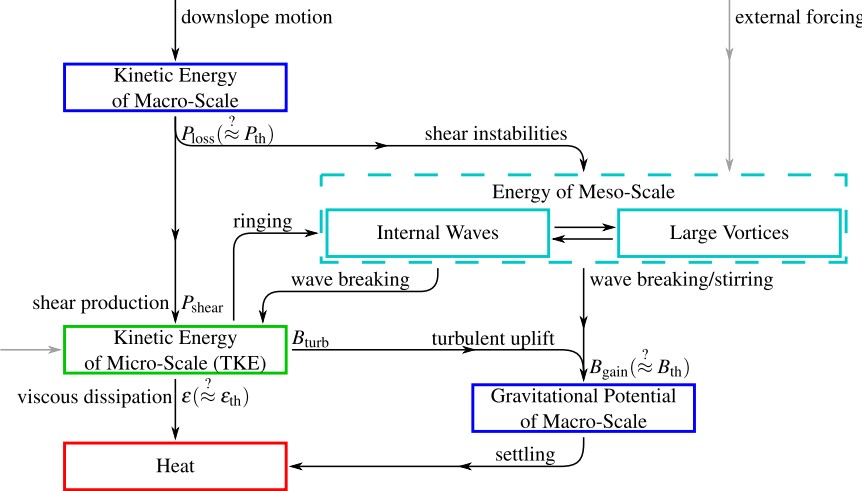

**Fig. 6 | Proposed internal energetics of gravity currents.** The internal stores of energy are shown by boxes, and the transfer of energy by black arrows: the energy lost at the tail of each arrow is equal to the energy gained at its tip. Grey arrows represent the input of energy from external forcing (energy could also be lost to these forces).

Therefore, a likely explanation (indeed the only one that remains) is that the top-hat model is wrong: the dynamics of the flow are not well represented if the information about the vertical variation of density and velocity are neglected. The origins of the top-hat model are in fluvial systems, where the log-law of the wall holds across a large portion of the flow. In this context, it is possible to calculate the shear production $P_{\mathrm{shear}}$ exactly[10,12,23,25,26], and the result is here denoted by $P_{\mathrm{f}} = u_\star^2 U$. A consequence of the driving density being uniform over the depth of the flow is that the top-hat model, in fluvial systems, gives exactly the same energy loss from the kinetic energy of the mean flow[30], which enables the top-hat models to accurately capture the flow energetics (up to internal waves and large-scale vortices). There is no reason to believe that the energetics of gravity currents are similarly well represented by top-hat models because the driving density varies over the depth. Experiments have shown that the shape factors resulting from the depth variation of density and velocity in laboratory currents differ from top-hat models by up to 40%[36]. In other settings, a strong density interface is generated around the maximum velocity, which is maintained through locally negative turbulent production driven by radiation stresses[39–41] analogous to atmospheric jets[55]. In either case, the loss of energy from the mean flow will be substantially different to the top-hat model. It is possible that $P_{\mathrm{loss}} > P_{\mathrm{th}}$, a large portion of this energy loss would need to go into the meso-scale structures because the mean-flow energy loss would be substantially larger than the shear production. In addition, the buoyancy production required by top-hat models is an upper bound, the sediment is assumed to be as high up as possible, and including the vertical variation of density would reduce this requirement so that $B_{\mathrm{gain}} < B_{\mathrm{th}}$. The reduction in the expected amount of energy passing through the TKE budget reduces the expected amount of energy loss to dissipation, meaning that a larger portion of the energy loss by the mean flow goes into particle uplift.

While investigations into the vertical structure of gravity currents have been conducted[36,56], the implications for the energetics remains an open problem. The momentum balance and sediment transport models may also need to be updated. Here, the vertical structure has already been incorporated in models[56], though it is not clear if this is sufficient to capture the effective force generated during the production of meso-scale structures, and the resulting dissipation-free uplift of particles.

The explanations can be summarised by writing

$$S = (P_{\mathrm{loss}} - P_{\mathrm{th}}) + (B_{\mathrm{th}} - B_{\mathrm{gain}}) + (\varepsilon_{\mathrm{th}} - \varepsilon), \qquad (8)$$

which summarises the effective extra energy available in a real current compared to the modelled current. Here, $P_{\mathrm{loss}} > P_{\mathrm{th}}$ due to the additional macro-scale kinetic energy lost to meso-scale internal waves and large vortices. This invalidates the long-standing Knapp–Bagnold hypothesis that all the energy lost by the macro-scale flow drives turbulence through shear production, and that turbulence is the only means of particle uplift. Note that the additional energy loss does not necessarily imply an energy depletion, real gravity currents may have a larger macro-scale kinetic energy budget than top-hat models due to the vertical variation of velocity. Additionally, $B_{\mathrm{gain}} < B_{\mathrm{th}}$, due to the lowering of the centre of mass when the vertical structure of density is captured[14], which reduces the gravitational potential that must be maintained. Finally, $\varepsilon < \varepsilon_{\mathrm{th}}$ because the TKE budget is lower, a large portion of the energy instead stored in the meso-scale structures. Thus, the presence of meso-scale structures increases the energetic efficiency of autosuspension.

To address the apparent missing energy in the models, future work must move beyond the approximations appropriate for open-channel and fluvial systems, and capture the complexity present in the structure and internal dynamics of gravity currents. The resulting additional capacity to support particles, that is the increased autosuspension capability, has numerous implications for environmental currents. The long run-out of turbidity currents has been a long standing enigma, and the results presented here show that the current is able to maintain a much higher sediment load than previously believed. This gives significantly more driving force on shallow slopes, and a much slower deposition rate of particles, which facilitates the transport of sediment to the distal parts of submarine systems. More broadly, particle-driven gravity currents are known to be highly destructive, with flows capable of causing immense damage. For the accurate prediction of gravity currents, this work shows that research focus is required on the dynamics of meso-scale energy exchange and balances, to be captured by the next generation of reduced order models.

## Methods

To evaluate the controls on the transport of sediment by turbidity currents, the dynamics of pseudo-steady state, and turbulent flows are examined. Both laboratory and real-world data are combined to cover a range of scales. To constrain the dynamics to pseudo-steady flows, only empirical data of continuous discharge or long-duration flows are considered. Rapidly varying flows, such as lock-exchange experiments or short-duration field observations, are omitted.

### Data analysis

Numerous studies report detailed vertical profiles of flow velocity and concentration data for gravity flows, including both sediment-laden turbidity currents and conservative composition-driven flows. The data for composition-driven flows are not used in the main analysis in this study, but instead to validate the applicability of the developed profile interpolation/extrapolation methodology (see Supplementary Figs. 3 and 4). Furthermore, three different datasets from direct measurements[57] of flows in the Congo canyon are added to this study to compare laboratory-scale and natural-scale turbidity currents. Since each study considered here uses different materials and methodologies, the comparison between datasets was based on consistent data interpolation and extrapolation methods to reconstruct the full vertical profiles of the flow (see Supplementary Note 2). In this study, the compiled data were categorised into two types: TYPE I) the gravity current data in which both vertical velocity and concentration profiles are measured, and TYPE II) gravity current data in which velocity profiles are available but vertical concentration profiles are not available.

Approximately two-thirds of compiled sources and the flume experiments in this study provide vertical profiles of both streamwise velocity and flow concentration (TYPE I: See Supplementary Table 1). For the remaining sources, either a part of or all of the reported experiments do not provide vertical concentration profiles (TYPE II). The depth-averaged flow concentration is estimated for those sources as follows. First, using the data that included both the concentration in the mixing tank and that measured in the flume (TYPE I), we construct an empirical relationship between these two quantities (see Supplementary Fig. 9). Then, the concentration in the flume (for those experiments that did not report it: TYPE II) is assumed to exactly satisfy the constructed relationship.

From the interpolated and extrapolated profiles, flow parameters are computed, including depth-averaged velocity, concentration, and flow depth (see Supplementary Note 1). The median particle size, $d_{50}$, is used to calculate the settling velocity of each experiment. For sand-sized particles, an empirical formula covering a combined viscous plus bluff-body drag law for natural irregular sand particles[58] is used, and for finer particles, Stokes' law is applied (see Supplementary Eq. (8)). For those data which used the sieving combined with hydrometer method (SHM) for the calculation of particle-size distribution, the reported median particle $d_{50}$ is amended based on the empirical relation between SHM and the laser diffraction method to avoid the

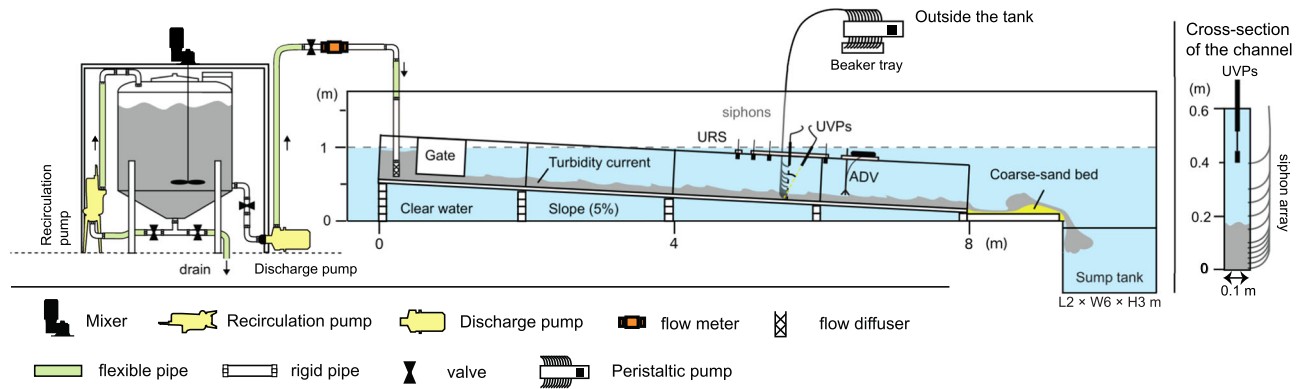

**Fig. 7 | Schematic illustration of the experimental flume.** The flow sampling location is at 4.7 m from the inlet (5.7 m downstream from the upstream end of the channel). The cross-section of the channel at the measurement location is depicted on the right side of the figure.

## Table 2 | Experimental conditions

| Experiment | $d_{50}$ (μm) | $\phi_0$ (vol.%) | $Q$ (l/s) | $D_{flow}$ (s) | $T_{ambient}$ | $T_{flow}$ (°C) | $\log_{10}$Re | Ri | $Re_p$ |
|---|---|---|---|---|---|---|---|---|---|
| 01 | 40 | 15.4 | 6 | 160 | 14.0 | 14.5 | 4.8 | 0.92 | 1.03 |
| 02 | 42 | 13.3 | 6 | 160 | 14.2 | 15.2 | 4.8 | 0.98 | 1.11 |
| 03 | 42 | 11.7 | 6 | 160 | 13.5 | 14.3 | 4.8 | 0.73 | 1.1 |
| 04 | 41 | 10.0 | 6 | 160 | 13.3 | 14.5 | 4.9 | 0.73 | 1.04 |
| 05 | 43 | 8.0 | 6 | 160 | 13.3 | 15.0 | 4.9 | 0.84 | 1.14 |
| 06 | 41 | 5.7 | 6 | 160 | 13.3 | 14.1 | 4.8 | 1.07 | 1.07 |
| 07 | 43 | 4.5 | 4 | 240 | 15.0 | 15.4 | 4.8 | 1.16 | 1.14 |
| 08 | 41 | 1.7 | 4 | 240 | – | – | 4.8 | 1.16 | 1.07 |
| 09 | 43 | 1.2 | 4 | 240 | – | – | 4.9 | 0.93 | 1.14 |
| 10 | 43 | 0.3 | 4 | 240 | 14.2 | 14.7 | 4.8 | 0.78 | 1.14 |
| 11 | 38 | 1.6 | 4 | 240 | 13.9 | 14.4 | 4.8 | 1.05 | 0.94 |
| 12 | 33 | 0.5 | 4 | 240 | 13.9 | 14.5 | 4.8 | 1.5 | 0.77 |
| 13 | 9 | 4.0 | 3.5 | 280 | 16.2 | 16.7 | 4.6 | 0.68 | 0.11 |
| 14 | 9 | 6.3 | 3.5 | 280 | 16.4 | 17.7 | 4.7 | 0.56 | 0.11 |
| 15 | 9 | 8.0 | 3.5 | 280 | 17.6 | 15.1 | 4.7 | 0.56 | 0.11 |
| 16 | 9 | 9.6 | 3.5 | 280 | 17.6 | 16.4 | 4.7 | 0.63 | 0.11 |
| 17 | 9 | 12.2 | 3.5 | 280 | 15.9 | 17.9 | 4.8 | 0.68 | 0.11 |

Median particle size ($d_{50}$), initial concentration in the mixing tank ($\phi_0$), discharge rate ($Q$), total flow duration ($D_{flow}$), the temperature of ambient water ($T_{ambient}$) and of the flow ($T_{flow}$), Reynolds number (Re), Richardson number (Ri), and particle Reynolds number ($Re_p$) are listed. $T_{ambient}$ and $T_{flow}$ are calculated by the acoustic Doppler velocimeter (ADV) as time-averaged values. The turbidity currents for the experimental runs are characterised by either glass beads (01–12) or kaolinite (13–17).

overestimation of clay fraction (see Supplementary Fig. 1 in online supplementary material).

### Flume experiments

The compiled dataset covers a wide range of flow concentrations ($10^{-2}$–$10^1$ vol.%), yet there are two data gaps, one of which is around 0.3–0.6 vol.% and the other is around 2–4 vol.% (see Supplementary Fig. 11). These gaps motivated new flume experiments, to better investigate the flow power balance of equilibrium turbidity currents. The experiments were conducted to study the sediment-load capacity of turbidity currents in an idealised channel[59] in the Total Environment Simulator, at the University of Hull. The main channel is 8 m long, 0.1 m wide, 0.6 m deep with a 5% slope, and it is submerged in a large water tank 12 m long, 6 m wide, 1 m deep filled with ambient water. At downstream, the channel is connected smoothly to a region of coarse sand of area 3 m × 3 m and 5 cm deep. A sump tank (2 m long × 6 m × 3 m) is located at the downstream end of the large tank, to minimise backwater effects[59].

Sediment-water mixtures are fed into the flume from a 1.0 m³ mixing tank (Fig. 7). Then, each experimental flow is discharged from

the flow diffuser pointed upstream to create a sediment-water cloud which generates a turbidity current by its negative buoyancy. The initial conditions for each run are set to fill the aforementioned two data gaps (Table 2). Velocity and density profiles are measured 4.7 m downstream from the inlet (Fig. 7). Velocity measurements are made using two Met-Flow Ultrasonic Velocity Profilers (UVPs), mounted at different angles: bed-normal and 30 degrees to the bed-normal angle. Suspended sediment samples are collected using a multi-channel peristaltic pump (Watson Marlow) connected to a 12-siphon array at a constant sampling rate (21 ml s⁻¹). Siphons are connected to a series of holes on the sidewall of the channel (0.7, 2.8, 4.7, 6.8, 9.0, 11.0, 16.0, 21.0, 26.0, 30.5, 35.7, and 40.5 cm above the bed, see Fig. 7) to minimise flow obstruction. To measure the aggradation rate, ultrasonic sensors and GoPro cameras are used near the measurement location (see Supplementary Note 3 for details).

### Data availability

The data points plotted in Figs. 1–3 are provided in the Source data file. The detailed reference and settings of the compiled sources, algorithm of flow interpolation/extrapolation and curve fittings are provided in

the Supplementary Information file. Source data are provided with this paper.

## Code availability

The codes used in this study are available in the following GitHub links: interpolation and extrapolation of flow profiles of gravity currents (https://github.com/SojiroFukuda/FlowProfiler) and the curve fitting in Fig. 3a (https://github.com/edskev/TC-not-rivers). The detailed formula and procedures of interpolation/extrapolation of profiles and the curve fittings implemented in these codes are also fully explained in the Supplementary Information (Note 1, 2 and 5).

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

## Acknowledgements
S.F., M.G.W.d.V., E.W.G.S., E.B., W.D.M. and R.M.D. were supported by the Turbidites Research Group, University of Leeds funded by AkerBP, CNOOC group, ConocoPhillips, Murphy Oil, OMV, Occidental Petroleum. R.M.D. was also supported by the UK Natural Environment Research Council [NE/S014535/1]. D.R.P., E.B., R.F. and X.W. were supported by the European Research Council under the European Union's Horizon 2020 research and innovation programme [grant 725955]. R.F. was also supported by the Leverhulme Trust, Leverhulme Early Career Researcher Fellowship [grant ECF-2020-679]. We thank Brendan J. Murphy, Fiona Chong, Anne Baar, and Steve M. Simmons for their assistance with the flume experiments at the Deep. We thank Steve M. Simmons for sharing the observation data from the Congo canyon. We are grateful for the UVP-DUO-MX equipment used herein, which was provided in collaboration with Met-Flow.

## Author contributions
S.F. undertook the data compilation and the statistical analyses. E.W.G.S., E.B., H.N., D.R.P. and R.M.D. contributed to interpretation of the results of the data compilation and the statistical analyses. E.W.G.S. developed the curve fitting methodology and conducted the curve fitting. S.F., M.G.W.d.V., E.B., R.F., X.W. and R.M.D. designed and conducted the flume experiments on which this study is based. All authors contributed to drafting and editing the manuscript.

## Competing interests
The authors declare no competing interests
