## [Peer Review File · Nature Communications]

Inadequacy of fluvial energetics for describing gravity current autosuspensionReviewers' Comments:

Reviewer #1:

Remarks to the Author:

The manuscripts re-analyzes existing turbidity current and open channel flow data, and presents new results from additional experiments, in order to compare the scaling behavior of both types of flows, with a focus on the dominant turbulent kinetic energy balance. Significant differences between the two types of flows are observed, which suggests that, in contrast to conventional wisdom, turbidity currents should not be viewed as being equivalent to "rivers on the seafloor." In particular, the present analysis suggests that there must be an additional source of turbulent kinetic energy for turbidity currents that increases their capacity to carry sediment, and that is not being accounted for when turbidity currents are treated as being equivalent to open channel flows.

The present investigations is comprehensive and has been carried out with care, and it has the potential to significantly alter our views as to the nature of turbidity currents. Hence I believe that the manuscript is appropriate for publication in Nat. Comm., provided that the following issues are addressed:

1. All dimensionless parameters need to be defined. For example, fig. 1 introduces the Froude number as well as the drag coefficient, but neither one is defined. Also, what is C_z , and for what purpose is it introduced?
2. The results shown in fig. 1 are not discussed or interpreted. What is the significance of the data clustering in different regions?
3. Caption of fig. 2: It says that ν is the dynamic fluid viscosity, but it should be the kinematic fluid viscosity.
4. In fig. 3a, only data from the Yellow River are shown. How close or different are these from corresponding data for other rivers?
5. Typo in line 124: compare Fig. 3c and 3c

Reviewer #2:

Remarks to the Author:

This is an interesting piece of work showing that an additional source of TKE would be necessary to explain long run distances of turbidity currents, and that open river models cannot sufficiently explain turbidity current dynamics. Unfortunately, this work does not provide the answer to what this additional TKE source would be, and thus does not explain the big question raised at the beginning of the manuscript - why turbidity currents traverse long distances over shallow slopes. Further, the point that turbidity currents behave different than river flows is not novel and somewhat obvious, and has been discussed in the literature for decades since Bagnold 1962. Although, this work is of high quality and represents a valuable contribution to the turbidity current community, it is very technical and highly specific, and I would suggest publication in a more specific journal. In its current form I find it a little difficult to see how it is novel and of broad enough interest to warrant publication in Nature Communications.

Reviewer #3:

Remarks to the Author:

I have now had the chance to read and review the article "Inadequacy of treating turbidity currents as

river analogues and implications for autosuspension criterion” by Fukuda et al. While the title of the manuscript might imply a broad look at a range of flow and sediment transport processes with a comparison of rivers and submarine channels, the focus is a bit narrower here, but a focus that is worth the effort directed at the problem addressed by Fukuda and colleagues. That focus is on the often-used Knapp-Bagnold (K-B) autosuspension criterion. This criterion states that the total production of turbulent kinetic energy (TKE) in a flow must exceed the sum of the work done to keep sediment in suspension and the viscous dissipation of energy in a flow if it is to keep sediment in suspension and thus be at or near equilibrium conditions necessary to transport sediment long distances in the ocean. This criterion was developed and has been verified extensively for open channel conditions and thus fluvial settings. A key focus of this study is whether a dimensionless flow power parameter that utilizes a TKE production parameter based on a logarithmic velocity profile scales with sediment concentration values in turbidity currents, and if so, what are the parameters that describe this scaling. Exploring this relationship is important as it is frequently used in numerical modeling studies of deep-sea settings. Given the limited data we have from field scale settings, numerical models that utilize this closure for sediment concentration are important for exploration of processes responsible for transporting and depositing the vast amounts of sediment carried by turbidity currents and the nutrients, carbon, and plastics also transported by these flows. Past modeling, study of this question also gives us a greater understanding of sediment transport in channelized flows in general.

To address this question, the authors compile and collect data on equilibrium flows at both laboratory and field scale. This is done with great care as to isolate data sets that truly are at equilibrium or close to equilibrium and to cover a wide range of conditions. In general, the authors find that the relationship between sediment concentration and dimensionless flow power for open channel conditions does not hold for turbidity currents. First, concentration to dimensionless flow power scaling in open channel flows is near linear for dilute flows. In contrast, this relationship is well non-linear for turbidity currents and data suggest this scaling is power-law in form with an exponent well below 1 (so you get less bang for your buck in the form of suspended sediment by increasing flow power in turbidity currents relative to open channel flows).

Next, using the very limited data available, the authors suggest that an estimation of TKE production through a logarithmic velocity profile closure drastically underpredicts measured TKE production. This point is explored within the discussion where the authors note several internal and external factors that might explain the missing TKE (i.e. TKE not produced by flow strain associated with the bottom flow boundary) necessary to keep sediment in suspension at equilibrium conditions in these flows. Unfortunately, this is largely an exploration of processes, leading to a closing statement that at present we just don't know where this production of TKE comes from, but it needs to come from somewhere.

I applaud the authors for constructing a well-executed study that is well reported. Strong intro, good data analysis methods, cutting edge theoretical treatment, and clearly engrained in the current literature of their community. Overall, I strongly support the publication of the work, though I do have a few issues I think could be better addressed in the manuscript. These are highlighted below.

First: Figure three is possibly the most important figure in the manuscript. However, it raises several questions for me. Some of these questions have to do with uncertainty of analysis, some link to a need for expanded explanation in the caption, some to do with interpretation. First, with regard to interpretation: in panel b you show a red dashed line as the relationship proposed by Bagnold. You also show a solid red line, which is your fit of dilute turbidity currents. I am struck by what appears to be two trends in the data. Many of the data points appear to follow the Bagnold relationship from very low to very high concentrations. The second trend is the one you highlight with the red solid line, which is described by a non-linear relationship with an exponent well less than 1. I am rather uncomfortable discounting either trend. While a further break down in the data exists in panels c-e, I am concerned that the x-axis range of Type I data is so limited and the data in Type II has so much scatter that neither really give me confidence that a non-linear relationship with an exponent less than 1 fully describes all data. Might you be able to explain away some of the scatter in the data that leaves these apparent two trends? For example, data from type II flows that have dimensionless stream powers around $2-3 \times 10^{-2}$, that also have concentrations around $1-2 \times 10^{-2}$ (why are these

concentrations so high relative to your trend). Or why do these data fall off your trend so much? Is there any justification you can give? Next, you state in lines 128-130 that some of your data come from experiments that use kaolinite and this might floc. But you discount this as a major problem. Can you inform the reader if the experiments that utilize kaolinite are those with high dimensionless flow power? Flocs are thought to occur in turbidity currents, producing flocs that can be >10x larger than the primary particles and with settling velocities that can be several orders of magnitude higher than the primary particle settling velocity. At present you reference a review paper from 2005 to argue that high shear flows will not have significant flocs within them,. But specifically for low Fr flows, recent work suggests this not to be the case. Specifically, work by Kyle Strom's group (Kuprenas, Rachel, Duc Tran, and Kyle Strom. "A shear-limited flocculation model for dynamically predicting average floc size." *Journal of Geophysical Research: Oceans* 123.9 (2018): 6736-6752.). Finally, for dilute turbidity currents there appears to be a gap in data between 10⁻¹ and 100 for the dimensionless flow power. Is there a reason for this data gap?

Next, given the scatter associated with the plots you present more information somewhere, even if just the supplemental, on the statistical methods employed to generate uncertainties on the fitting coefficients in table 2? I argue that properly documenting methods to determine uncertainties are critical for us evaluating the confidence in which we use relationships reported in these types of studies.

With regards to the discussion, it would be great if different internal/external sources of TKE production (e.g. channel bends, Coriolis Force, internal waves) could have some numbers on the possible TKE they might provide? To what degree could they close the gap in the missing TKE noted? I know from a paper I published in 2011 (Straub, Kyle M., et al. "Quantifying the influence of channel sinuosity on the depositional mechanics of channelized turbidity currents: A laboratory study." *Marine and Petroleum Geology* 28.3 (2011): 744-760.) that channel bends can induce strong mixing in a turbidity current, redistributing mass from close to the bed to higher up in the flow. Any quantification on the magnitudes of these TKE sources would be helpful. Further, I was hoping the authors could comment on the importance and possible underappreciation of TKE produced from bed roughness. Might commonly used Cd values to estimate u* be too low? I understand that this is a slight aside as your measurements in figure 3 use full profiles of velocity and as such u* can be directly calculated. However, how we apply this to models in which we do not have velocity profiles is also important. The final point I will make: The title highlights an "Inadequacy of treating turbidity currents as river analogues". While I don't disagree with this premise, I will say the following. Most of the community already knows that there are significant differences between rivers and turbidity currents. This has been highlighted in concentration profiles that do not follow Rouse profiles, differences in the helical flow structure around channel bends, structure of velocity fields, etc. Yes, this is a new set of observations that highlights differences. However, the use of the analogue has been a beneficial starting point for the community. One that allowed us to leverage the vast amount of work done on the dynamics of open channel flow, much of which is pertinent to understanding the fluid dynamics and transport capacity of turbidity currents, even if the dynamics vary around the margins (or maybe well past the margins as is argued here). I just worry that the title and introduction of this ms sets up a strawman for the authors to knock over. I think the quality of the work and the importance of the question asked does not require this strawman.

Again, great study!

Best,

Kyle M. Straub

The authors would like to thank the reviewers for their detailed and thorough review of the manuscript. We have made amendments to the text and figures to address all of the reviewers' comments. There are two versions of manuscript: 1) the revised manuscript with track changes highlighted in yellow, 2) the revised manuscript without track-change highlighting. In this document, the authors' response to comments is highlighted in blue. Where used in the authors' response, line numbers (LN) refer to the revised manuscript with track changes (version 1). Below we summarise main changes to manuscript before providing detailed response to comments.

The original manuscript, analysis and discussion focused on gravity current dynamics in terms of a dimensionless log-law TKE production term, $P_f/N_f = u_*^2 U / gRh w_s$ (P/B' in the original version), as the approximation of total-shear production of TKE P_{shear}/N_f based on fluvial theory. We concluded that the additional TKE source would be either i) self-induced or ii) via an environment-induced production mechanism, speculating it is likely to be the self-induced production because environment-induced production cannot explain the non-linear trend in laboratory-scale data.

In the revised manuscript we have simplified the introduction of P_f/N_f and used the opportunity to expand the analysis to incorporate the full production term $P_{th} = u_*^2 U + e_w U^3$ which describes the total energy loss of a mean flow in the top-hat gravity current model (see LN 50–85 in the Introduction, LN 122–129, 144–148 in the Results and the Discussion section). This is the natural generalisation of the log-law production in fluvial systems to gravity currents, because it modifies the production by the effect the upper shear layer. Whether this top-hat model adequately represents the flow is now a central question of the paper. Consequently, data filtering based on the top-hat model power balance between driving force and the frictional forces is omitted in the revised manuscript.

The principal conclusion remains the same: a non-linear flow power correlation is demonstrated between the flow concentration, Φ , and the dimensionless top-hat flow power term, P_{th}/N_{th} (Fig. 3c and Table 1-iv), contrary to all extant models. The inclusion of entrainment in P_{th}/N_{th} improves the quality of the non-linear fit, strengthening the result. The secondary conclusion also stands: there is insufficient energy in the model to support the sediment loads observed in laboratory and environmental currents (Fig. 4a and b). Thus, we make the stronger statement that the top-hat model energetics do not represent real gravity currents (Discussion section).

We welcomed reviewer requests for additional detail on the unbalanced energy budget, thus we added a new discussion and figure to breakdown gravity current energetics (Fig. 6 and LN 184–254). This concludes both that P_{th} is not a good proxy for, and indeed overestimates, shear production, and that meso-scale energy exchange, such as internal waves and coherent structures, are a primary mechanism for uplifting the suspended sediments. Because those flow structures can uplift sediment with/without exchanging the kinetic energy to TKE, the assumption that the energy lost by the large-scale flow equals turbulent shear production is invalid, LN 236-239.

Here, $P_{loss} > P_{th}$ due to the additional macro-scale kinetic energy lost to meso-scale internal waves and large vortices. The substantive conclusion of the manuscript is that this invalidates the long-standing Knapp-Bagnold hypothesis that all the energy lost by the macro-scale flow drives turbulence through shear production, and that turbulence is the only means of particle uplift.

Response to Reviewer 1

The manuscripts re-analyzes existing turbidity current and open channel flow data, and presents new results from additional experiments, in order to compare the scaling behavior of both types of flows, with a focus on the dominant turbulent kinetic energy balance. Significant differences between the two types of flows are observed, which suggests that, in contrast to conventional wisdom, turbidity currents should not be viewed as being equivalent to “rivers on the seafloor.” In particular, the present analysis suggests that there must be an additional source of turbulent kinetic energy for turbidity currents that increases their capacity to carry sediment, and that is not being accounted for when turbidity currents are treated as being equivalent to open channel flows.

The present investigations is comprehensive and has been carried out with care, and it has the potential to significantly alter our views as to the nature of turbidity currents. Hence I believe that the manuscript is appropriate for publication in Nat. Comm., provided that the following issues are addressed:

We thank the reviewer for taking the time to review our manuscript, and for their constructive comments. We have endeavoured to address them, which has improved the quality and clarity of the manuscript.

1. All dimensionless parameters need to be defined. For example, fig. 1 introduces the Froude number as well as the drag coefficient, but neither one is defined. Also, what is C_z , and for what purpose is it introduced?

The Chezy number and Froude number are manipulations of the defined drag coefficient, C_D , and the bulk Richardson number, Ri . Thus, the figure axes of Figure 1 are revised to simplify the manuscript; i.e. respectively C_D and Ri . The caption is amended as needed.

2. The results shown in fig. 1 are not discussed or interpreted. What is the significance of the data clustering in different regions?

We thank the reviewer for highlighting this omission. The manuscript has been updated to note that, as summarised in Figure 1, the range of compiled sources covers both supercritical to subcritical flow conditions and small to large drag coefficients.

LN 97–98:

➤ *“Figure 1 highlights that this dataset uniformly spans a wide range of flow states, from subcritical to supercritical with small to large drag coefficients.”*

3. Caption of fig. 2: It says that ν is the dynamic fluid viscosity, but it should be the kinematic fluid viscosity.

This error has been amended as requested.

4. In fig. 3a, only data from the Yellow River are shown. How close or different are these from corresponding data for other rivers?

A wide range of additional field and laboratory open-channel data from different sources are now included in Figure 3a. The added data show excellent agreement with the Yellow River data. The detailed references of fluvial data is added in the supplementary material (Supplementary Table 2, LN 238–243 in Supplementary Information).

5. Typo in line 124: compare Fig. 3c and 3c

The majority of the Results section is modified based on the new Figure 3. This figure has been modified not only to include additional fluvial data, but also data for pyroclastic flows, and the sub-figures and axis have also been changed in line with the substantially amended text. The sentence with this typo is no longer a part of the manuscript.

Response to Reviewer 2

This is an interesting piece of work showing that an additional source of TKE would be necessary to explain long run distances of turbidity currents, and that open river models cannot sufficiently explain turbidity current dynamics.

We thank the reviewer for the positive review of our manuscript.

Unfortunately, this work does not provide the answer to what this additional TKE source would be, and thus does not explain the big question raised at the beginning of the manuscript - why turbidity currents traverse long distances over shallow slopes.

We thank the reviewer for the opportunity to provide more details on explanations for autosuspension. We have taken this opportunity to amend the discussion to motivate how suspension mechanics, determining autosuspension, in gravity currents should be revised

The conclusion of the non-linear correlation between P_{th} and B_{th} , see following comment, has significant implication for autosuspension as: i) a higher driving force on shallower slopes is anticipated as more sediment can be suspended at lower P_{th} ; ii) the shallower correlation suggests that flows are less sensitive (in terms of deposition-erosion) to changes in slope. These arguments are detailed on LN 130–136:

- *"Critically the non-linear relationship results in dilute gravity currents being able to maintain a higher suspended sediment concentration versus fluvial flows of an equivalent dimensionless flow power (Fig. 3c). Previously unrecognised, this has the potential to explain autosuspension in long-runout systems. The correlation suggests that when a dilute gravity current accelerates, it is not as erosive as a fluvial system, and similarly, when a gravity current decelerates, deposition is more limited. This implies that the suspended-load of gravity currents is significantly underestimated, i.e. providing more motive force on shallower slopes, and is less sensitive to changes in flow power than has previously been assumed based on the use of fluvial analogues."*

A further, critical, conclusion of the non-linear relationship is that there exists a significant energy deficit: i.e. the autosuspension enigma for gravity currents. In response we have amended the manuscript to provide a careful review of the energetics of gravity currents.

LN 174–177:

- *"The energy balance of gravity currents required for autosuspension cannot be explained without an additional energy source, i.e. $S > 0$ in Equation (5) and Figure 4. Crucially, if the energetic mechanisms were the same for gravity currents and fluvial flows, then gravity currents would be substantially more dilute, cf. Figure 3b–e. Therefore, to explain autosuspension, mechanisms for particle uplift must be present that are absent, or of negligible importance, in fluvial flows."*

LN 178–181:

- *"It is plausible that the shear production, Equation (2), predicted by top-hat models is less than the actual production in real flows... However, the limited data available suggest that the directly computed shear production term, P_{shear} , is substantially lower than the 'top-hat' energy loss term, P_{th} ."*

LN 197–199:

- *"The internal shear generated by the macro-scale flow also directly energises turbulent vortices through shear production. The meso-scale flow structures are able to generate gravitational potential directly, by stirring the flow, and through wave breaking which also generates TKE."*

LN 212–213:

- *"Therefore, a likely explanation (indeed the only one that remains) is that the top-hat model is wrong: the dynamics of the flow are not well represented if the information about the vertical variation of density and velocity are neglected."*

In summary this review concludes that energy must be balanced by mixing at the meso-scale, as summarised in a new Figure – Figure 6.

Further, the point that turbidity currents behave different than river flows is not novel and somewhat obvious, and has been discussed in the literature for decades since Bagnold 1962.

Bagnold 1962 frames turbidity currents as a river analogue explicitly, in terms of the transfer of kinetic energy of the flow to turbulent kinetic energy at the micro-scale. All system-scale gravity current models since, which by scale constraint adopt some form of the top-hat model, have made this assumption. Top-hat gravity current models make a minor modification to the Bagnold energy balance to take ambient fluid entrainment effects into account. We have amended the manuscript to highlight these points on LN 75–77:

- *“The dimensionless flow power inherent to top-hat models of gravity currents can be derived from the kinetic energy conservation equation of the mean flow, which is only modified by the presence of entrainment”.*

Moreover, we include a revised Figure 3, contrasting the non-dimensional flow power term from Parker’s model P_{th}/N_{th} to show that the production term in top-hat gravity current models exhibits a non-linear correlation with concentration Φ (i.e. normalised buoyancy production). This non-linear correlation is further highlighted through a new table, Table 1, stating correlation and goodness of fit.

The non-linear correlation between energy production and analysis of available power versus work done keeping sediment in suspension is the novel contribution from the manuscript and is substantially different from extant energetic balance models currently used.

LN 19–21:

- *“Contrary to current theory, buoyancy production is shown to have a non-linear dependence on available flow power, indicating an underestimation of the total kinetic energy lost from the mean flow.”*

However, in recognition that other differences between rivers and turbidity currents have been identified previously, we have amended the title to be more specific about the new contribution we make:

- *“Inadequacy of fluvial energetics for describing gravity current autosuspension”*

Although, this work is of high quality and represents a valuable contribution to the turbidity current community, it is very technical and highly specific, and I would suggest publication in a more specific journal. In its current form I find it a little difficult to see how it is novel and of broad enough interest to warrant publication in Nature Communications.

We have extensively reviewed, amended and updated the manuscript to improve accessibility and clarity, including modifications to the:

Manuscript

- Simplify the technical aspects of the text, where possible, whilst retaining detail.
- Revise and simplify mathematical notation used throughout and reducing the number of equations to essential statements.
- Incorporate additional data and studies of other forms of particulate-laden gravity currents, pyroclastic density currents, which strengthen the principle conclusions.

Abstract

- Highlight the link and importance to generic gravity currents, ubiquitous in a wide range of natural and industrial settings.

Introduction

- Provide detailed motivation for resolving the dynamics of gravity currents.
- Explain why turbidity currents provide an excellent case study of gravity currents.

Discussion

- Provision of a new summary diagram (Figure 6) that clearly highlights that the energy balance is explained by meso-scale energetics not captured in extant system scale ‘top-hat’

models. The diagram clarifies the notation used and functions as a quick reference for what each of the symbols represent.

To evidence the broader interest and application of the research we have modified the manuscript, data and analysis to extend beyond turbidity currents. Based on the Reviewer's comments we have included dynamically similar pyroclastic density current data, which matches to our analysis of turbidity current data. Thus, we are grateful to the reviewer for challenging us to develop broader interest in the manuscript, which has strengthened our results and the implications of the work.

Further, we broadened the abstract and introduction to gravity currents in general, respectively

LN14–22:

- *“Gravity currents, such as sediment-laden turbidity currents, are ubiquitous natural flows that are driven by a density difference. ... Here, empirical research from different types of particle-driven gravity currents is integrated with our experimental data, to show that material transport is fundamentally different from fluvial systems. ... A revised energy budget directly implies that the mixing efficiency of gravity currents is enhanced.”*

LN 25–37:

- *“Gravity currents are a broad class of flows with a wide range of environmental applications, including terrestrial cold fronts and submarine thermohaline currents ... However, despite its importance the mechanisms that enable autosuspension are poorly understood because the kinetic energy of the flow is consumed to maintain the particles in suspension, and uplift them during turbulent mixing with the environment, which ultimately stalls the flow.”*

LN 92–93:

- *“A review of the energy deficit implies that particulate transport in gravity currents is driven by mixing at scales larger than that of TKE.”*

We also added a discussion of the observed non-linear trend of pyroclastic flows

LN 127–129, 136–138:

- *“Moreover, the turbidity current data (Fig. 3d) show almost identical non-linear dependency to the pyroclastic density current data (Fig. 3e), suggesting that the non-linear dependency is universal to all types of gravity currents.”*
- *“Since the limited super-dilute pyroclastic density current dataset also exhibits a similar non-linear trend to turbidity currents (Fig. 3e), it is likely that the fluvial-based or top-hat gravity current models are a poor approximation not only for turbidity currents but also for particle-driven gravity currents in general.”*

Finally, the discussion section has been modified extensively to address the energy balance of gravity currents in general rather than being specific to turbidity currents.

LN 149–254:

- *“The fundamental differences between fluvial flows and dilute gravity currents (turbidity currents and pyroclastic density currents) documented above raise the following questions: ... More broadly, particle-driven gravity currents are known to be highly destructive, with flows capable of causing immense damage. For the accurate prediction of gravity currents, this work shows that research focus is required on the dynamics of meso-scale energy exchange and balances, to be captured by the next generation of reduced order models.”*

In summary, we believe that this manuscript has a broad interest to anybody interested in dynamics of gravity currents, and of mixing in stratified turbulent flow in general. Both of these occur in, and have significant relevance to, a wide range of natural and industrial settings and therefore we argue that the manuscript, presenting a substantial and novel change in understanding of the energetics of gravity currents is appropriate for publication

in Nature Communications, which has a recent track-record of publishing high quality research in turbidity currents (see, e.g., Hughes-Clarke, 2016; Paull et al., 2018).

Response to Reviewer 3

I have now had the chance to read and review the article “Inadequacy of treating turbidity currents as river analogues and implications for autosuspension criterion” by Fukuda et al. While the title of the manuscript might imply a broad look at a range of flow and sediment transport processes with a comparison of rivers and submarine channels, the focus is a bit narrower here, but a focus that is worth the effort directed at the problem addressed by Fukuda and colleagues.

That focus is on the often-used Knapp-Bagnold (K-B) autosuspension criterion. This criterion states that the total production of turbulent kinetic energy (TKE) in a flow must exceed the sum of the work done to keep sediment in suspension and the viscous dissipation of energy in a flow if it is to keep sediment in suspension and thus be at or near equilibrium conditions necessary to transport sediment long distances in the ocean. This criterion was developed and has been verified extensively for open channel conditions and thus fluvial settings. A key focus of this study is whether a dimensionless flow power parameter that utilizes a TKE production parameter based on a logarithmic velocity profile scales with sediment concentration values in turbidity currents, and if so, what are the parameters that describe this scaling. Exploring this relationship is important as it is frequently used in numerical modeling studies of deep-sea settings. Given the limited data we have from field scale settings, numerical models that utilize this closure for sediment concentration are important for exploration of processes responsible for transporting and depositing the vast amounts of sediment carried by turbidity currents and the nutrients, carbon, and plastics also transported by these flows. Past modeling, study of this question also gives us a greater understanding of sediment transport in channelized flows in general.

To address this question, the authors compile and collect data on equilibrium flows at both laboratory and field scale. This is done with great care as to isolate data sets that truly are at equilibrium or close to equilibrium and to cover a wide range of conditions. In general, the authors find that the relationship between sediment concentration and dimensionless flow power for open channel conditions does not hold for turbidity currents. First, concentration to dimensionless flow power scaling in open channel flows is near linear for dilute flows. In contrast, this relationship is well non-linear for turbidity currents and data suggest this scaling is power-law in form with an exponent well below 1 (so you get less bang for your buck in the form of suspended sediment by increasing flow power in turbidity currents relative to open channel flows).

Next, using the very limited data available, the authors suggest that an estimation of TKE production through a logarithmic velocity profile closure drastically underpredicts measured TKE production. This point is explored within the discussion where the authors note several internal and external factors that might explain the missing TKE (i.e. TKE not produced by flow strain associated with the bottom flow boundary) necessary to keep sediment in suspension at equilibrium conditions in these flows.

We thank the reviewer for taking the time to conduct a thorough and substantial review of our manuscript, and for their constructive comments. We have endeavoured to address them below. We believe that they have improved the quality and clarity of the manuscript.

Unfortunately, this is largely an exploration of processes, leading to a closing statement that at present we just don't know where this production of TKE comes from, but it needs to come from somewhere.

I applaud the authors for constructing a well-executed study that is well reported. Strong intro, good data analysis methods, cutting edge theoretical treatment, and clearly engrained in the current literature of their community. Overall, I strongly support the publication of the work, though I do have a few issues I think could be better addressed in the manuscript. These are highlighted below.

We are grateful of the reviewer's supportive comments.

Developed through the Editor's and all the Reviewer's feedback and comments, for which we are grateful, the manuscript has been significantly modified to advance the discussion of autosuspension interlinked to the mixing processes in gravity currents.

The reviewer notes a shortcoming in our identification of where the TKE comes from. The paper has been revised, and new analysis reveals that the correct explanation is that the dynamics circumvent TKE altogether. Figure 5 shows that the energy going to TKE through

shear production in real flows is even less than the energy made available by the macro-scale flow in the top-hat model, which is in turn less than the energy required to suspend particles. Instead, meso-scale mixing processes are generated, and mix the sediment throughout the flow without ever generating TKE. The total energy transferred from the macro-scale flow to both turbulence and meso-scale structures is larger than predicted by the top-hat model. This highlights the shortcomings of extant 'top-hat' models for system-scale gravity current modelling, in general, and is positioned to stimulate discussion within the gravity currents community.

The non-linear dependency of dimensionless flow power against the flow concentration also demonstrates the capability of flows to support much more material than previously postulated. This implies there is a greater driving force for a given flow power and thus slope. Thus, we revise the manuscript to highlight that

LN 248–251:

- *“The long run-out of turbidity currents has been a long standing enigma, and the results presented here show that the current is able to maintain a much higher sediment load than previously believed. This gives significantly more driving force on shallow slopes, and a much slower deposition rate of particles, which facilitates the transport of sediment to the distal parts of submarine systems.”*

A consequence of the revision of the manuscript was to consider the correlation between Parker's 'top-hat' energy production and buoyancy production. It is important to note that the classical force balance of Parker does not hold a priori. Thus, no direct connection can be made between enhanced sediment load and bypass slopes. However, it should be expected that with equilibrium concentration and thus Richardson number increasing the equilibrium bypass slope can decrease.

First: Figure three is possibly the most important figure in the manuscript. However, it raises several questions for me. Some of these questions have to do with uncertainty of analysis, some link to a need for expanded explanation in the caption, some to do with interpretation. First, with regard to interpretation: in panel b you show a red dashed line as the relationship proposed by Bagnold. You also show a solid red line, which is your fit of dilute turbidity currents. I am struck by what appears to be two trends in the data. Many of the data points appear to follow the Bagnold relationship from very low to very high concentrations. The second trend is the one you highlight with the red solid line, which is described by a non-linear relationship with an exponent well less than 1. I am rather uncomfortable discounting either trend.

To clarify how poor the Bagnold relationship is as the predictor of dilute turbidity currents, we have made several alterations to the text and figures.

We now directly compare the top-hat model, which is the natural generalisation of the log-law for fluvial systems, modified by the effects of the entrainment in the upper shear layer. This removes the appearance of a linear (Bagnold) fit in the data, and the non-linear trend is now visually very clear. We also added data from two more turbidity current experiments and from dilute pyroclastic density currents data in the analysis (Fig. 1–3 in the main text and Supplementary Table 1).

Correlation coefficients as well of goodness of fit for different subsets of data and models were evaluated and are compared in a new Table 1. Specifically, when considering concentration as a function of the dimensionless 'top-hat' flow power for turbidity current and pyroclastic density current data (Fig. 3c)

i) a linear fit yields an R^2 of 0.34 (Table 1-i)

ii) the optimal non-linear power-law, a power of 0.49, yields an $R^2=0.72$ (Table 1-iv)

For gravity currents, the non-linear correlation between P_{th} and B_{th} is an inescapable conclusion.

We also improved the quality and consistency of the compiled dataset. Data in all laboratory-scale experiments are calculated as the mean profile of each run, such that one

data point from one run. However, each point of natural-scale turbidity current data in the original version of the manuscript was extracted as 500 second averages of velocity and concentration profiles. Consequently, there were more than 100 data points from each event (Event 01, 04 and 05 last more than 5 days), with inherent variation from tidal forcing and waxing and waning of the flow. For consistency, we changed the data compilation method for Congo's flows and extracted a single mean profile from each event, averaging selected windows (Supplementary Fig. 5–8). With vertical profiles of Congo's flows recognised to vary in time, a two-day window average (Supplementary Fig. 5) is now used to calculate the depth-averaged parameters from each event.

Supplementary information LN 166–173:

- *“Firstly, the moving averages of velocity and concentration time series are calculated from each event (Supplementary Fig. 5). Then, the depth-averaged flow parameters, (flow velocity and concentration), are calculated from each averaged profile (Supplementary Fig. 6–8) ... The velocity relative to the time-averaged background velocity is extracted as the flow velocity measurements.”*

While a further break down in the data exists in panels c-e, I am concerned that the x-axis range of Type I data is so limited and the data in Type II has so much scatter that neither really give me confidence that a non-linear relationship with an exponent less than 1 fully describes all data. Might you be able to explain away some of the scatter in the data that leaves these apparent two trends? For example, data from type II flows that have dimensionless stream powers around $2-3 \times 10^{-2}$, that also have concentrations around $1-2 \times 10^{-2}$ (why are these concentrations so high relative to your trend). Or why do these data fall off your trend so much? Is there any justification you can give?

We thank the reviewer for highlighting this point via the data point examples. The data spread, in the revised manuscript Figure 3b, arises because P_f and B_f are very poor predictors for gravity currents. It is seen that by including entrainment that the experimental data aligns better. This is seen to occur predominately in the transitional zone when increased entrainment rates of supercritical flows enhances buoyancy production quicker than 'top-hat' energy production. This narrows the spread of data, compare the transitional zone of Figure 3b and 3c.

However, it is noted that empirical entrainment models used are only designed for dilute flows. Entrainment in more concentrated flows may differ, due to the role of particle-particle interaction and enhanced effective flow viscosity.

LN 144–148:

- *“Moreover, whilst the increase in P_{th}/N_{th} of gravity currents (Fig. 3c) follows the trend of fluvial data in the transitional regime it is based on an empirical water entrainment function, e_w . The empirical water entrainment function has been developed for dilute currents, thus it is expected the values of P_{th}/N_{th} in the transitional regime have some inherent error. However, this does not impact the primary findings of the non-linear correlation in the dilute regime.”*

Next, you state in lines 128-130 that some of your data come from experiments that use kaolinite and this might floc. But you discount this as a major problem. Can you inform the reader if the experiments that utilize kaolinite are those with high dimensionless flow power? Flocs are thought to occur in turbidity currents, producing flocs that can be $>10x$ larger than the primary particles and with settling velocities that can be several orders of magnitude higher than the primary particle settling velocity. At present you reference a review paper from 2005 to argue that high shear flows will not have significant flocs within them,. But specifically for low Fr flows, recent work suggests this not to be the case. Specifically, work by Kyle Strom's group (Kuprenas, Rachel, Duc Tran, and Kyle Strom. "A shear-limited flocculation model for dynamically predicting average floc size." *Journal of Geophysical Research: Oceans* 123.9 (2018): 6736-6752.).

We thank the reviewer for the valuable insights related to the flocculation. In the revised Fig. 3 the nonlinear trend fits all of the data well, with the kaolinite experiments matching the trend just as well as the non-cohesive experiments. Additionally, the kaolinite experiments (Tesaker 1969) exhibit one of the highest Froude numbers in our dataset (the

densimetric Froude number Fr 1.5–2.5), and thus the Kyle Strom's group's work for the low Fr flows in tidal or river setting which the reviewer provided is not likely to be the case for the kaolinite turbidity currents in this study. However, we modify the text to note LN 139–144:

- *“It is noted that a limited proportion of the data (see Supplementary Table 1) use cohesive material (kaolinite). It is plausible that flocculation of cohesive particles may occur, increasing particle settling velocity, and decreasing dimensionless flow power. However, the growth of flocs is limited by shear rates and their size decreases as the flow increases. Therefore, in strongly sheared gravity current experiments, the potential development of flocs and underestimation of settling velocity is likely to be limited. Further, it would not change the observation that the relationship between concentration and dimensionless flow power is non-linear (Fig. 3b–e, Table 1 ii) and iv).”*

Finally, for dilute turbidity currents there appears to be a gap in data between 10⁻¹ and 10⁰ for the dimensionless flow power. Is there a reason for this data gap?

There is a limited availability of published long-duration experiments of turbidity currents with flow concentration 0.5%. We added a new figure in the supplementary material (Supplementary Fig. 12) to show the available data range of the compiled data in terms of the flow power plot, in which all data points without the data reduction in the main text are plotted. The data gap is not due to the data reduction, but because of the lack of experimental study in this regime. Our laboratory experiments were planned to fill this data gap, but resulted in lower dimensionless flow power than expected, and so fell outside the gap.

We have added a paragraph in the supplementary material discussing the limited data range.

Supplementary information LN 229–232:

- *“There are few data points in the dilute regime around $10^{-1} < P_f / N_f < 10^0$. This is not because the data is excluded due to the introduced equilibrium criteria (Fig. 2 in the main text) but simply the long-duration experiments of turbidity currents with flow concentration, $\Phi \sim 0.5\%$ are limited. Our flume experiments partially fill this data gap by adding three runs within the target concentration range.”*

With regards to the discussion, it would be great if different internal/external sources of TKE production (e.g. channel bends, Coriolis Force, internal waves) could have some numbers on the possible TKE they might provide? To what degree could they close the gap in the missing TKE noted? I know from a paper I published in 2011 (Straub, Kyle M., et al. "Quantifying the influence of channel sinuosity on the depositional mechanics of channelized turbidity currents: A laboratory study." *Marine and Petroleum Geology* 28.3 (2011): 744-760.) that channel bends can induce strong mixing in a turbidity current, redistributing mass from close to the bed to higher up in the flow. Any quantification on the magnitudes of these TKE sources would be helpful.

We thank the reviewer for a very good question. From the revised analysis, we concluded that the environment-induced TKE production cannot be the extra source of missing energy, thus quantifying the potential energy from environmental forcing in the natural-scale event is not the main focus of the revised manuscript. Instead, the missing energy is argued to be generated by the non-top-hat structure of the flow, and transferred to the internal waves and coherent structures, which then transfer the energy to gravitational potential of suspended particles. Although top-hat gravity current model provides a measure of this total energy loss of a mean flow, the approximation makes the prediction of the models very poor, and thus the modelled production does not represent the energy lost from the largest flow scale, which highlights the open question of how to model of energy balance in gravity currents.

Having said that, direct observations of natural-scale events are still limited, and our manuscript does not deny the existence of environmental forcing which feeds a substantial amount of TKE to the natural-scale flow. Therefore, to quantify the potential TKE from those system-scale events are also important. To do so, the measurements of instantaneous 3-

dimensional flow velocity field both from laboratory- and natural-scale turbidity currents are required, however, such data is very limited, and technically challenging. The available data is primarily for straight channel flows, where indeed we use the revised discussion to note LN 218–221:

- *“There is no reason to believe that the energetics of gravity currents are similarly well represented by top-hat models because the driving density varies over the depth. Experiments have shown that the shape factors resulting from the depth variation of density and velocity in laboratory currents differ from top-hat models by up to 40%.”*

However, there is no work where the relationship between secondary cell strength and the mixing efficiency are quantified. Yet, as the implication from Straub et al. (2010), the impact of these events against the material-transport efficiency of turbidity currents are potentially significant. We recognise this in the revised discussion

LN 196–199:

- *“Alternatively, the mean flow energy may be used to stimulate large vortices, for example, secondary flow circulation[Straub 2010] ... The meso-scale flow structures are able to generate gravitational potential directly, by stirring the flow[Straub 2010] and through wave breaking which also generates TKE.”*

This problem is not only important for the turbidity currents community, but also for the other large-scale density currents phenomenon such as pyroclastic flows and snow avalanches.

Further, I was hoping the authors could comment on the importance and possible underappreciation of TKE produced from bed roughness. Might commonly used C_D values to estimate u^* be too low? I understand that this is a slight aside as your measurements in figure 3 use full profiles of velocity and as such u^* can be directly calculated. However, how we apply this to models in which we do not have velocity profiles is also important.

We thank the reviewer for the valuable insight. We added an additional paragraph in the supplementary material discussing the correlation (see Supplementary Fig. 10). The results imply that the correlation between the commonly used C_D and the estimated u_* is poor. We do not include this discussion in the main text, to avoid distraction from the main message, but is an important area which deserves attention.

Supplementary Information LN 186–194

- *“In this study the drag coefficient, C_D is estimated from the velocity gradient (Eq. 6–7)... While the flows with the low top-hat drag coefficient ($Rg\Phi hS/U^2 \leq 0.05$) show a relatively poor correlation with the velocity-profile-based drag coefficient.”*

The final point I will make: The title highlights an “Inadequacy of treating turbidity currents as river analogues”. While I don’t disagree with this premise, I will say the following. Most of the community already knows that there are significant differences between rivers and turbidity currents. This has been highlighted in concentration profiles that do not follow Rouse profiles, differences in the helical flow structure around channel bends, structure of velocity fields, etc. Yes, this is a new set of observations that highlights differences. However, the use of the analogue has been a beneficial starting point for the community. One that allowed us to leverage the vast amount of work done on the dynamics of open channel flow, much of which is pertinent to understanding the fluid dynamics and transport capacity of turbidity currents, even if the dynamics vary around the margins (or maybe well past the margins as is argued here). I just worry that the title and introduction of this ms sets up a strawman for the authors to knock over. I think the quality of the work and the importance of the question asked does not require this strawman.

Again, great study!

Best,

Kyle M. Straub

We thank the reviewer for the warm compliments and valuable suggestion.

Bagnold 1962 set turbidity currents as river analogue explicitly. The implications of this have been in all models of gravity currents. (The drag coefficient and erosion have been modelled as fluvial system as well). Although the upper shear is taken into account in some

of the unstratified numerical models of turbidity currents ('top-hat' models, e.g. Parker et al., 1986), those common models are also based on fluvial theories.

Our results indicate that these top-hat gravity current models are very poor approximation of the energy balance of gravity currents, which again highlight the fundamental difference in energy transfer mechanics between fluvial and turbidity currents which has not been considered to date.

We have modified the title as follows:

Inadequacy of fluvial energetics for describing gravity current autosuspension

Reviewers' Comments:

Reviewer #3:

Remarks to the Author:

I have now had the chance to read and review the manuscript: "Inadequacy of fluvial energetics for describing gravity current autosuspension" by Fukuda et al. This is a rereview of a ms. The current ms is substantially different from the first submission and the authors took on most of the suggested edits provided by myself and the other reviewers. I will be happy to see this in press and recommend that the ms be published in Nature Comms.

As the authors detail, the central goal of the document is an evaluation of the Knapp-Bagnold (K-B) autosuspension criterion for density currents. This criterion was initially developed for fluvial systems and is commonly used in numerical modeling efforts for density currents and in particular turbidity currents. However, a central challenge that we face as the deep water sediment transport community is to understand how turbidity currents can run out for extremely long distances. Application of the K-B criterion in numerical models generally results in conditions that cannot produce long run out currents. This ms does an excellent job of collecting data from fluvial systems and density currents (both turbidity currents and pyroclastic flows) to compare suspension capacity (depth averaged concentration values) to top hat models that estimate the production and loss of turbulent kinetic energy. The authors show that turbidity currents are able to maintain concentrations higher than would be predicted by fluvial scaling.

A possible short-coming of the paper is that the authors highlight a problem, but don't solve it. I take a different read. Sometimes a scientific community needs to have a problem clearly laid out in a ms to instigate studies that will solve the problem. Here the authors highlight additional energy sources that might aid sediment suspension in turbidity currents that are currently being ignored and basically sets forth a challenge to the community to quantify these sources and add them to models. I applaud this call and think this work will be influential in setting the tone for future studies. Thank you for giving me the chance to review the work.

Best,
Kyle M. Straub

The authors would like to thank the reviewers for their detailed and thorough review of the manuscript. In this document, the authors' response to comments is highlighted in blue.

Response to Reviewer 3

I have now had the chance to read and review the manuscript: "Inadequacy of fluvial energetics for describing gravity current autosuspension" by Fukuda et al. This is a rereview of a ms. The current ms is substantially different from the first submission and the authors took on most of the suggested edits provided by myself and the other reviewers. I will be happy to see this in press and recommend that the ms be published in Nature Comms.

As the authors detail, the central goal of the document is an evaluation of the Knapp-Bagnold (K-B) autosuspension criterion for density currents. This criterion was initially developed for fluvial systems and is commonly used in numerical modeling efforts for density currents and in particular turbidity currents. However, a central challenge that we face as the deep water sediment transport community is to understand how turbidity currents can run out for extremely long distances. Application of the K-B criterion in numerical models generally results in conditions that cannot produce long run out currents.

This ms does an excellent job of collecting data from fluvial systems and density currents (both turbidity currents and pyroclastic flows) to compare suspension capacity (depth averaged concentration values) to top hat models that estimate the production and loss of turbulent kinetic energy. The authors show that turbidity currents are able to maintain concentrations higher than would be predicted by fluvial scaling.

A possible short-coming of the paper is that the authors highlight a problem, but don't solve it. I take a different read. Sometimes a scientific community needs to have a problem clearly laid out in a ms to instigate studies that will solve the problem. Here the authors highlight additional energy sources that might aid sediment suspension in turbidity currents that are currently being ignored and basically sets forth a challenge to the community to quantify these sources and add them to models. I applaud this call and think this work will be influential in setting the tone for future studies. Thank you for giving me the chance to review the work.

Best,
Kyle M. Straub

We thank the reviewer for taking the second time to conduct a thorough review of our manuscript, and for their very supportive comments.